# Common Era sea-level budgets along the U.S. Atlantic coast

Jennifer S. Walker [1,2✉], Robert E. Kopp [1,2], Timothy A. Shaw [3], Niamh Cahill [4], Nicole S. Khan[5], Donald C. Barber [6], Erica L. Ashe [1,2], Matthew J. Brain[7], Jennifer L. Clear[8], D. Reide Corbett [9] & Benjamin P. Horton [3,10]

Sea-level budgets account for the contributions of processes driving sea-level change, but are predominantly focused on global-mean sea level and limited to the 20th and 21st centuries. Here we estimate site-specific sea-level budgets along the U.S. Atlantic coast during the Common Era (0–2000 CE) by separating relative sea-level (RSL) records into process-related signals on different spatial scales. Regional-scale, temporally linear processes driven by glacial isostatic adjustment dominate RSL change and exhibit a spatial gradient, with fastest rates of rise in southern New Jersey (1.6 ± 0.02 mm yr$^{-1}$). Regional and local, temporally non-linear processes, such as ocean/atmosphere dynamics and groundwater withdrawal, contributed between −0.3 and 0.4 mm yr$^{-1}$ over centennial timescales. The most significant change in the budgets is the increasing influence of the common global signal due to ice melt and thermal expansion since 1800 CE, which became a dominant contributor to RSL with a 20th century rate of 1.3 ± 0.1 mm yr$^{-1}$.

[1] Department of Earth and Planetary Sciences, Rutgers University, New Brunswick, NJ, USA. [2] Rutgers Institute of Earth, Ocean and Atmospheric Sciences, Rutgers University, New Brunswick, NJ, USA. [3] Earth Observatory of Singapore, Nanyang Technological University, Singapore, Singapore. [4] Department of Mathematics and Statistics, Maynooth University, Maynooth, Ireland. [5] Department of Earth Sciences and Swire Marine Institute, The University of Hong Kong, Hong Kong, Hong Kong. [6] Departments of Environmental Studies and Geology, Bryn Mawr College, Bryn Mawr, PA, USA. [7] Department of Geography, Durham University, Durham, UK. [8] Department of Geography and Environmental Science, Liverpool Hope University, Liverpool, UK. [9] Department of Coastal Studies, East Carolina University, Greenville, NC, USA. [10] Asian School of the Environment, Nanyang Technological University, Singapore, Singapore. ✉email: walker@marine.rutgers.edu

Sea-level budget assessments quantify the different physical processes contributing to sea-level change[1]. Since 1993, sea-level budget assessments generally combine satellite altimetry estimates of total global-mean sea-level (GMSL) rise and, more recently, Argo float-derived estimates of global-mean thermosteric sea-level rise with satellite gravimetric measurements of barystatic contributions (i.e., from land-ice and land water) and process model-derived information[2]. For example, the GMSL budget (1993–2018) indicates the thermosteric contribution (1.3 mm yr$^{-1}$) was dominant with additional contributions from glaciers and ice sheets (~0.7 mm yr$^{-1}$ each)[3]. The 20th century GMSL budget has been estimated with the additional aid of tide-gauge data and direct measurements of changing mass balance of glaciers and ice sheets with a total rate of rise of 1.2 ± 0.2 mm yr$^{-1}$ from 1901–1990[4]. Studies on preinstrumental timescales are limited to single time slices, such as quantifying the ice sheet contributions to the GMSL lowstand during the Last Glacial Maximum[5]. A further limitation of sea-level budgets is the paucity of regional and local relative sea-level change assessments[6]. Relative sea level (RSL) differs from GMSL because of driving processes such as glacial isostatic adjustment (GIA); ocean dynamic sea-level change; gravitational, rotational, and deformational (GRD) responses to barystatic changes; tectonics; and sediment compaction[7,8]. The driving processes are spatially variable and cause RSL change to vary in rate and magnitude among regions[9].

Sea-level budgets for the Common Era (0–2000 CE) are unknown, but proxy RSL reconstructions have extended the instrumental record back before the 19th century[10] and have improved understanding of magnitudes, rates, and driving processes of regional sea-level change at centennial to multidecadal timescales[11]. For example, along the U.S. Atlantic coast, GIA has been a significant driving factor in RSL rise through the Common Era, creating a spatially variable signal due to the region's proximity to the former Laurentide Ice Sheet[12,13]. Larger uncertainty is associated with the remaining RSL processes occurring on different spatial scales, including common global signals driven by thermosteric and barystatic changes; regional signals such as ocean dynamic sea-level change and GRD changes; and local site-specific signals such as tidal range change and sediment compaction.

Here, we estimate site-specific Common Era sea-level budgets along the U.S. Atlantic coast. We complete a new high-resolution (decimeter vertical scale, decadal temporal scale) RSL record in northern New Jersey (Fig. 1), filling in a spatial data gap between RSL records in southern New Jersey[14] and New York City[15]. Integrating this new record into an updated global database of instrumental and proxy sea-level records[11,16] (Supplemental Data), we use a spatiotemporal empirical hierarchical model[11,17] to examine magnitudes and rates of Common Era RSL in the northern New Jersey record and five other published records along the U.S. Atlantic coast (Fig. 1a). The high concentration of these high-resolution RSL reconstructions over a 700 km stretch of coastline allows the records to be decomposed into process-related signals on different spatial scales, assisting in interpretation of processes that drive both spatial and temporal patterns of sea-level changes.

## Results

**Common Era sea-level trends**. To examine regional RSL trends over the Common Era for the U.S. Atlantic coast, we compare 100-year average rates from Cheesequake State Park in northern New Jersey to published proxy-based sea-level reconstructions (Fig. 1a).

The last millennium reconstruction of RSL from northern New Jersey (Fig. 1d, Supplementary Figs. 1–9) reveals that RSL rose continuously from 1000–2000 CE at an average rate of 1.5 ± 0.2 mm yr$^{-1}$ (2σ). Over the preindustrial Common Era from 0–1700 CE, RSL in northern New Jersey rose at a rate of 1.3 ± 0.2 mm yr$^{-1}$. This rate is consistent with the spatial gradient from the other U.S. Atlantic coast sites, with the fastest rates of rise occurring in southern New Jersey with 1.6 ± 0.1 mm yr$^{-1}$ at Leeds Point (LP) and 1.5 ± 0.1 mm yr$^{-1}$ at Cape May Courthouse (CMC) (Fig. 2). In New York City, there were slightly slower rates of rise of 1.2 ± 0.1 mm yr$^{-1}$, and the slowest rates of rise occurred in Connecticut and North Carolina with 1.0 ± 0.1 and 1.1 ± 0.1 mm yr$^{-1}$, respectively. The spatial gradient of Common Era RSL rates is a result of each location's position relative to the time-evolving position of the former Laurentide Ice Sheet[13].

The reconstructions over the last 300 years show elevated RSL rates at all six sites, ranging from 1.7 ± 0.3 mm yr$^{-1}$ in Connecticut to 2.5 ± 0.3 mm yr$^{-1}$ in southern New Jersey (CMC). It is extremely likely (probability $P \geq 0.98$) that the average rate of rise from 1700–2000 CE at all six sites was faster than during any preceding 300-year period during the preindustrial Common Era (0–1700 CE). Furthermore, it is virtually certain ($P > 0.99$) that the 20th century rate of rise at all six sites was faster than during any preceding century in the Common Era.

The updated global database of instrumental and proxy records of the Common Era (Fig. 3a) illustrates a global-scale component, or globally uniform signal, of RSL trends similar to previous results[11,16]. Global sea level gradually rose from 0 to 500 CE at a rate of 0.1 ± 0.1 mm yr$^{-1}$ (1σ), but then gradually fell from 500 to 1300 CE at a rate of −0.1 ± 0.1 mm yr$^{-1}$ (Fig. 3b). It then rose from 1300 to 1600 CE at a rate of 0.1 ± 0.2 mm yr$^{-1}$ and fell again from 1600 to 1800 CE at a rate of −0.1 ± 0.2 mm yr$^{-1}$. Global sea-level rise beginning in the 19th century reached an average rate of 1.3 ± 0.1 mm yr$^{-1}$ in the 20th century. It is virtually certain ($P > 0.999$) that the global 20th century rate of rise was faster than during any preceding century during the Common Era.

**Sea-level budgets: global component**. The global component of RSL, identified as a signal common to all of the records in the Common Era database, is relatively small (i.e., centimeter-scale) and fairly stable without any consistent periodicity at multidecadal to multicentennial timescales until the onset of modern rates of rise (Fig. 3). Rates of the global component fluctuated between −0.2 and 0.2 mm yr$^{-1}$ from 0 to 1800 CE, with an increasing rate of rise since 1800 CE, reaching 1.3 ± 0.1 mm yr$^{-1}$ in the 20th century (Fig. 4, Table 1).

Our global sea-level estimate using the globally uniform signal is consistent with GMSL budgets for the 20th century, which found total rates of rise of 1.2 ± 0.2 mm yr$^{-1}$ (1901–1990) (90% confidence interval)[18] and 1.56 ± 0.33 mm yr$^{-1}$ (1900–2018) (90% confidence interval)[19]. Prior to the 20th century, there is no evidence for global sea-level rate changes associated with the time period of the Medieval Climate Anomaly; however, there is a negative global contribution during the latter part of the Little Ice Age (Fig. 5a), which coincides with a period of decreased global air and sea surface temperatures and the most extensive glacial advances in the Common Era (Fig. 5b, c)[20,21]. The enhanced rates of rise and greatest contribution of the global component that followed in the 20th century are caused primarily by increased ocean mass and volume from glacier and ice sheet melt and thermal expansion on a global scale in response to greenhouse forcing of warming sea surface and surface air temperatures (Fig. 5b, c)[22]. The increasing influence of the global component is the most significant change in the sea-level budgets at all six sites

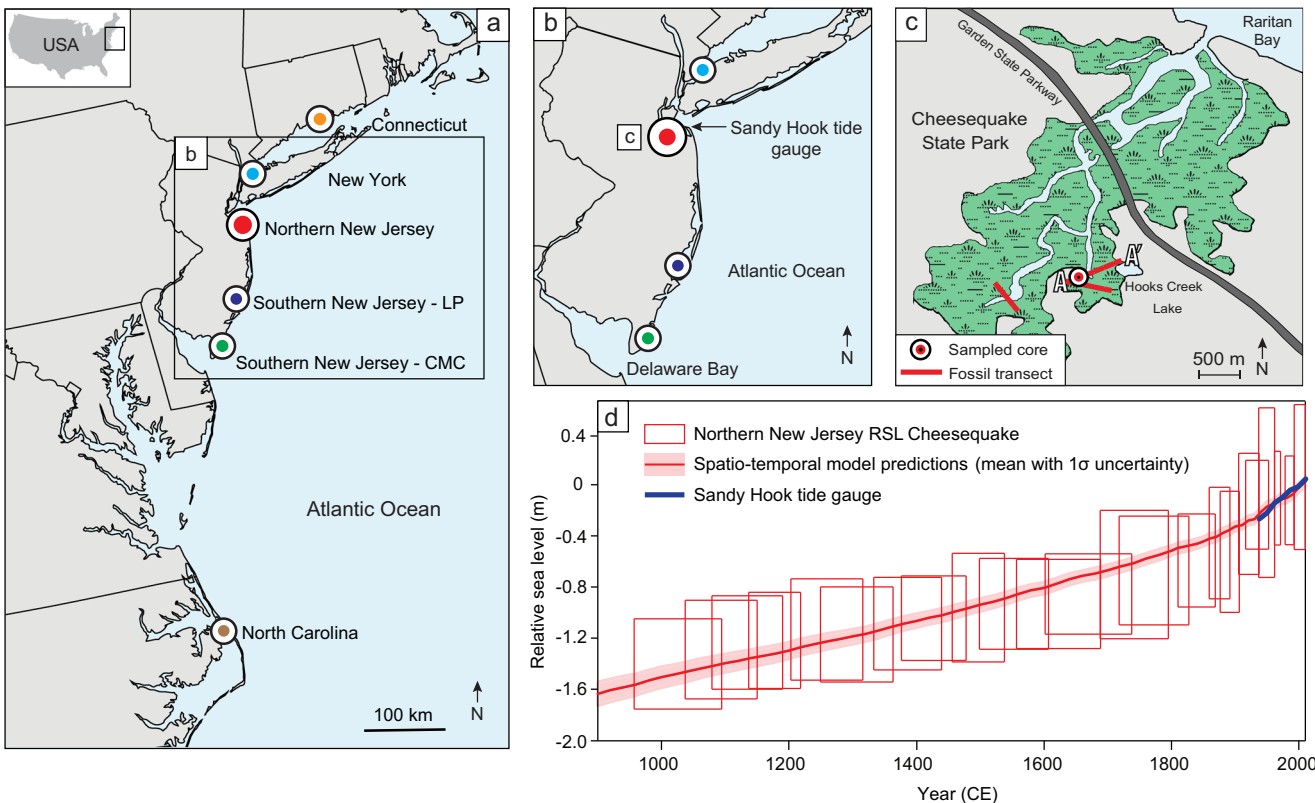

**Fig. 1 Location of relative sea-level (RSL) reconstructions on the U.S. Atlantic coast. a** The six sites used in analysis from north to south are: Connecticut[60], New York City[15], northern New Jersey (this study), southern New Jersey (Leeds Point (LP) and Cape May Courthouse (CMC)[14]), and North Carolina[45]. **b** Map of New Jersey with location of new northern New Jersey record and Sandy Hook tide gauge. **c** Salt-marsh study site at Cheesequake State Park in northern New Jersey off of Raritan Bay where the new RSL record was produced. Fossil core transect locations shown, including location of the sampled core for detailed analysis. **d** RSL record from northern New Jersey with spatiotemporal model predictions (mean with 1σ uncertainty) and decadal-average RSL measurements from the Sandy Hook tide gauge. Boxes represent the vertical RSL (1σ) and chronological (2σ) uncertainty for each data point.

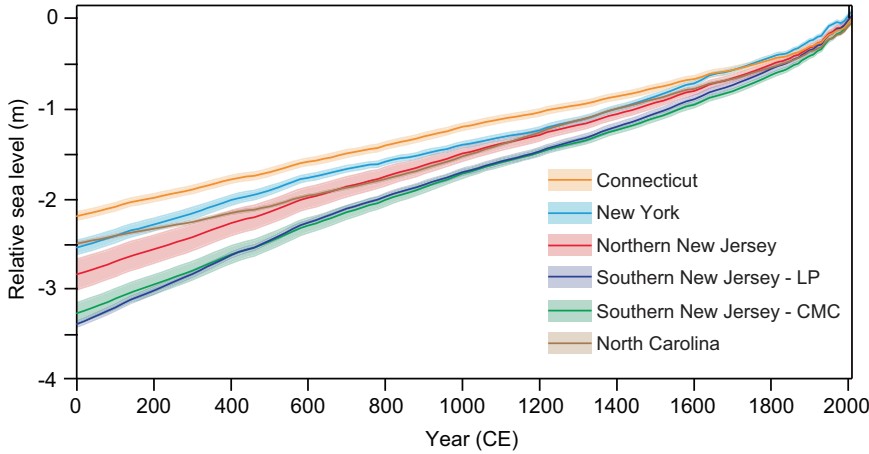

**Fig. 2 Common Era relative sea-level model predictions for U.S. Atlantic coast sites.** See Fig. 1a for site locations. Model predictions are the mean with 1σ uncertainty. LP Leeds Point. CMC Cape May Courthouse.

through the Common Era (Fig. 4). By 1950, global sea-level change was responsible for 36–50% of RSL change at each site.

**Sea-level budgets: regional linear component**. The dominant processes driving RSL change at U.S. Atlantic coast sites are regional-scale and temporally linear, contributing 2.1–3.2 m of rise over the Common Era (Fig. 4). In northern New Jersey, the linear rate contribution is $1.3 \pm 0.08$ mm yr$^{-1}$ (Table 1). The linear component exhibits a spatially variable signal, with the greatest

contributions in southern New Jersey of $1.6 \pm 0.02$ mm yr$^{-1}$ (LP) and $1.5 \pm 0.05$ mm yr$^{-1}$ (CMC) and smaller contributions to the north and south, with the smallest in Connecticut at $1.0 \pm 0.03$ mm yr$^{-1}$. The spatial variability of the dominant linear contribution is consistent with the effects of GIA. As the Laurentide ice sheet retreated, the peripheral forebulge began to collapse, causing land subsidence and maximum rates of RSL rise in the mid-Atlantic region (e.g., New Jersey)[12]. A similar spatial pattern and magnitude of land subsidence, likewise attributed to GIA, has been described

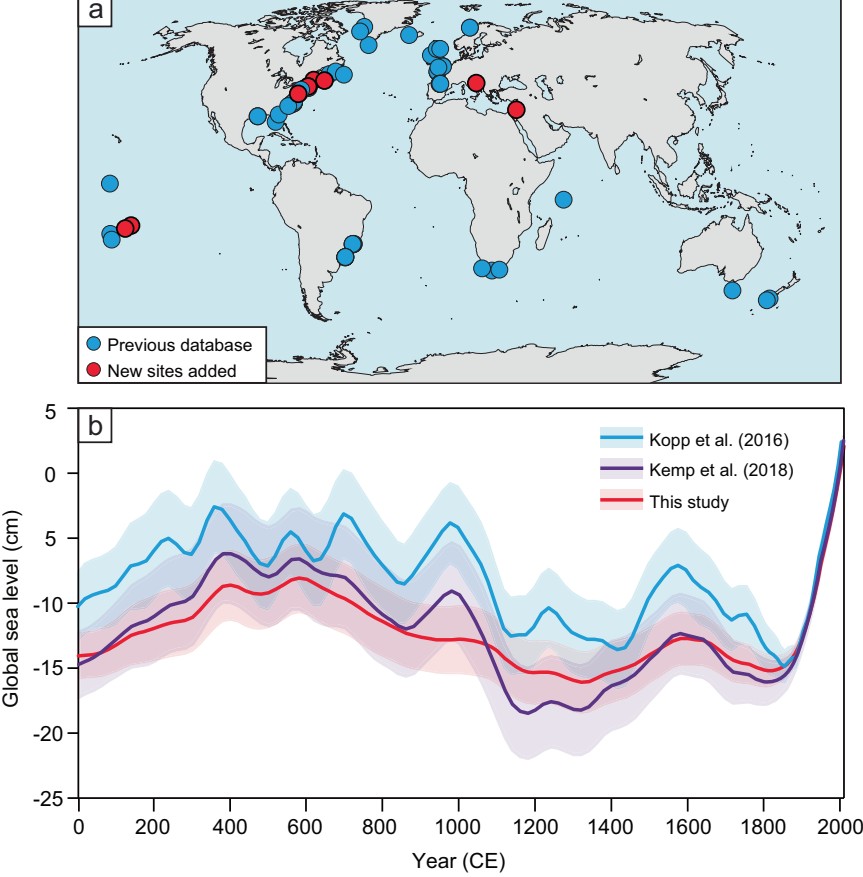

**Fig. 3 Global sea-level contribution. a** Proxy sea-level records in the Common Era sea-level database, where new sites updated from Kemp et al.[16] are shown in red. **b** Reconstructed global sea level from this study compared to the results from Kopp et al.[11] and Kemp et al.[16]. Model predictions are the mean with 1σ uncertainty.

on the U.S. Atlantic coast from the late Holocene through the 20th century[23]. Coastal plain locations (e.g., New Jersey) have also experienced higher rates of RSL rise than locations underlain by bedrock (e.g., New York City) due to the natural compaction of unconsolidated Late Quaternary coastal plain sediments, with an average 20th century compaction rate of $0.16 \, \mathrm{mm \, yr^{-1}}$ (90% confidence interval, $0.06–0.32 \, \mathrm{mm \, yr^{-1}}$)[24]. Therefore, coastal plain subsidence may also contribute to the greater linear contribution in New Jersey. At all sites, the temporally linear processes provide the dominant contribution to sea-level rise throughout the Common Era, while the nonlinear signals from global, regional, and local-scale processes yield comparatively smaller contributions (Supplementary Figs. 10, 11). Although the combined nonlinear signal varies on multidecadal to multicentennial timescales, we do not identify any consistent periodicity, nor any rapid, large magnitude changes in rate as have been described recently[25]. Instead the linear contribution clearly dominates RSL rise through the Common Era until it is matched or surpassed by the global contribution in the 20th century.

**Sea-level budgets: regional nonlinear component.** Regional-scale nonlinear contributions are similar across U.S. Atlantic coast sites and have an amplitude of <15 cm through the Common Era (Fig. 4, Supplementary Fig. 12). For example, in northern New Jersey, regional nonlinear processes contributed a negative influence from 0–500 CE, followed by a period of stability from 500–1000 CE. A second interval of negative influence occurred from ~1000 CE until 1600 CE, after which the regional nonlinear contribution became

positive and increased until present. The contribution of the regional nonlinear component to northern New Jersey RSL fluctuated between $-0.2$ and $0.2 \, \mathrm{mm \, yr^{-1}}$ during the preindustrial Common Era before increasing to a rate of $0.4 \pm 0.2 \, \mathrm{mm \, yr^{-1}}$ in the 20th century (Table 1). Across the U.S. Atlantic coast sites, the regional nonlinear contribution is nearly identical from Connecticut to southern New Jersey and differs only slightly in North Carolina, where rates fluctuated between $-0.3$ and $0.2 \, \mathrm{mm \, yr^{-1}}$ during the preindustrial Common Era before increasing to a rate of $0.4 \pm 0.2 \, \mathrm{mm \, yr^{-1}}$ in the 20th century (Fig. 4).

The regional nonlinear contribution is likely explained by a combination of changing physical processes through the Common Era. The negative regional nonlinear influence from 0–500 CE could be driven by regional-scale steric effects[26] associated with long-term cooling broadly in the Northern Hemisphere and within the North Atlantic following early to mid-Holocene maxima[16,27]. The stabilization of the regional nonlinear contribution from 500–1000 CE could indicate a reduction in long-term cooling or, alternatively, additional processes driving an opposing influence on regional sea-level trends to offset the cooling effects. Changes in atmosphere and ocean circulation that have altered prevailing winds and ocean currents can drive regional sea-level changes[16]. For example, changes in the strength of the Atlantic Meridional Overturning Circulation (AMOC) and the Gulf Stream can cause sea-level changes on centimeter to decimeter scales on the U.S. Atlantic coast[28]. Additionally, proxy reconstructions of the North Atlantic Oscillation (NAO) provide evidence for changing atmospheric circulation over the Common Era, which could manifest in

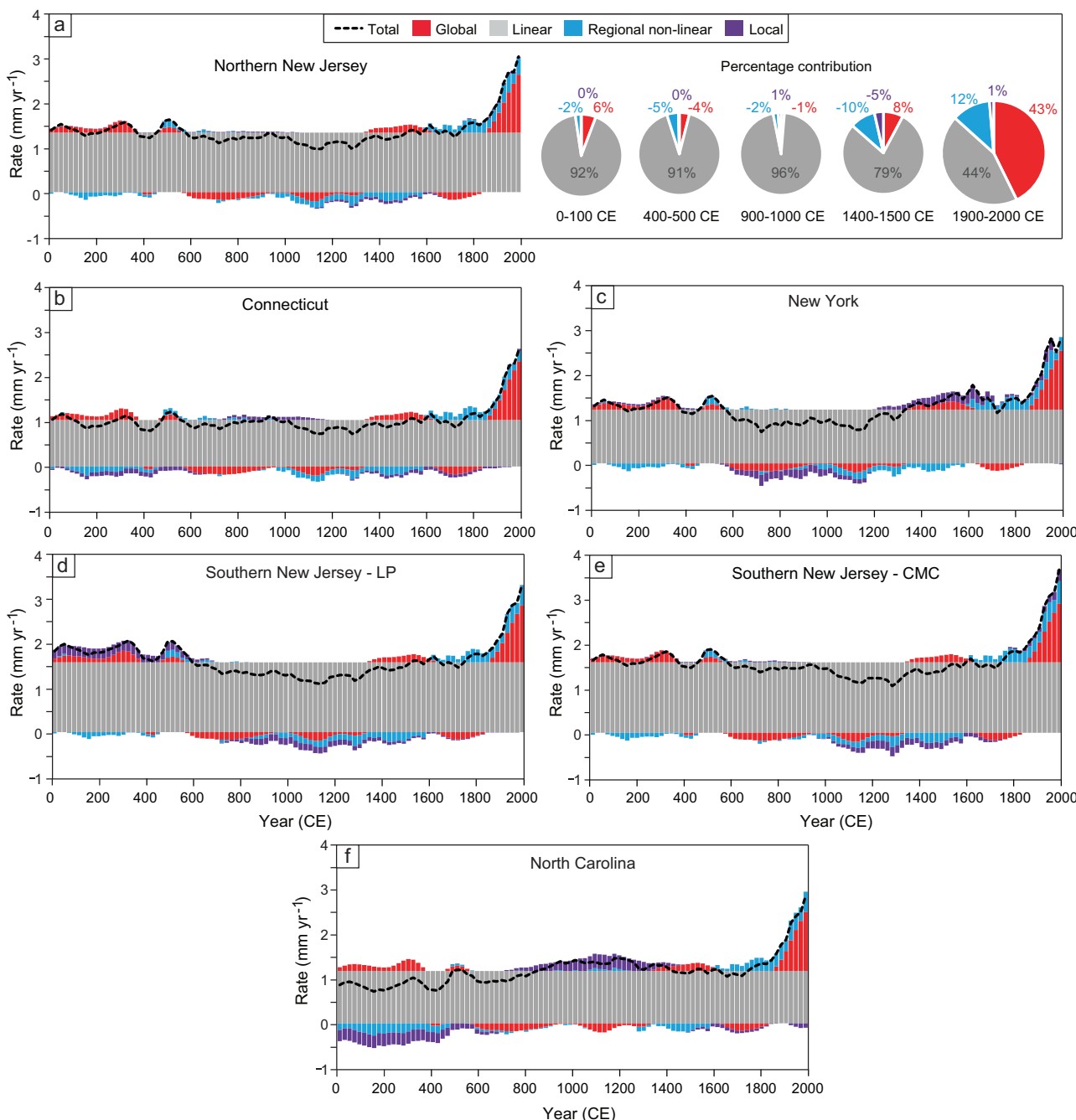

**Fig. 4 Common Era sea-level budgets.** Budgets are separated into global, linear, regional nonlinear, and local components for **a** northern New Jersey; **b** Connecticut; **c** New York; **d** southern New Jersey—Leeds Point (LP); **e** southern New Jersey—Cape May Courthouse (CMC); **f** North Carolina. Total rates for each site are indicated by dashed black line. Percentage contributions of each budget component are also shown for northern New Jersey for five 100-year time intervals through the Common Era.

centimeter-scale changes in regional sea level[29]. The shift from a negative to positive regional nonlinear contribution around 1600 CE (Fig. 5a) may be explained by broad climate transitions in the North Atlantic. Beginning around 1400 CE, the NAO changed from a sustained positive phase to a negative phase and AMOC experienced a weakening (Fig. 5d)[30]. If a decrease of 1 Sv in overturning transport can cause an increase in sea level of ~1.5 cm on the northeast U.S. Atlantic coast[31], then the positive contributions from the regional nonlinear component could at least be partially explained by an AMOC weakening over the last 500 years. There is additional evidence for a weakening AMOC over the industrial period (Fig. 5d)[32], which may also correspond

to the positive regional nonlinear contribution specifically in the last century.

Additionally, the evolving mass of ice sheets could contribute to regional-scale nonlinear RSL trends in the mid-Atlantic over the Common Era. However, the behaviour of the Antarctic Ice Sheet is poorly constrained over this time period[33] and while the Greenland Ice Sheet is better constrained, its potential influence on the regional nonlinear trends is uncertain[34]. The Greenland Ice Sheet may have advanced and reached a peak in mass by the end of the Little Ice Age and subsequently began melting and losing mass to the global ocean[35]. However, other studies show minimal variability in ice-mass loss over the Common Era (Fig. 5c)[36],

**Table 1 Common Era sea-level budget for northern New Jersey for five 100-year time intervals.**

| Time Interval | Average rate (mm/yr) | | | | |
| --- | --- | --- | --- | --- | --- |
| | Global | Regional linear | Regional nonlinear | Local | Total |
| 0–100 CE | 0.08 ± 0.20 | 1.34 ± 0.08 | −0.03 ± 0.26 | 0 ± 0.25 | 1.39 ± 0.40 |
| 400–500 CE | −0.06 ± 0.20 | 1.34 ± 0.08 | −0.07 ± 0.26 | 0 ± 0.25 | 1.22 ± 0.40 |
| 900–1000 CE | −0.02 ± 0.20 | 1.34 ± 0.08 | −0.03 ± 0.26 | 0.02 ± 0.25 | 1.31 ± 0.40 |
| 1400–1500 CE | 0.13 ± 0.17 | 1.34 ± 0.08 | −0.17 ± 0.26 | −0.06 ± 0.25 | 1.24 ± 0.38 |
| 1900–2000 CE | 1.30 ± 0.06 | 1.34 ± 0.08 | 0.37 ± 0.15 | 0.04 ± 0.25 | 3.05 ± 0.29 |

Rates are the mean with 1σ uncertainty. Processes are separated by spatial components.
Controlling processes:
Global component: ocean density changes; land-ice-mass changes.
Regional linear component: glacial isostatic adjustment; long-wavelength sediment compaction.
Regional nonlinear component: atmosphere/ocean dynamics; gravitational, rotational, and deformational effects of land-ice changes.
Local component: short-wavelength sediment compaction; tidal range change; anthropogenic groundwater withdrawal (20th century).

which is supported by the fact that nonlinear changes in mass of the Greenland Ice Sheet should produce a spatially variable fingerprint along the U.S. Atlantic coast, increasing from north to south, and such a fingerprint is not discernible in the six sites analysed here. Additionally, the effects of the ice sheet may be too small or overprinted by other processes that exhibit a greater nonlinear signal (e.g., steric effects, ocean circulation changes) to be detected in the regional-scale nonlinear trends[16]. As multiple processes may act simultaneously and with opposing influences on regional sea-level trends, the methods used here cannot fully distinguish and quantify the relative magnitude of each individual process comprising the regional nonlinear contribution.

**Sea-level budgets: local component.** The local-scale contribution comprises those trends unique to each individual site that are not observed over larger spatial scales. The local component is spatially and temporally variable; however, its interpretation is limited by the extent and resolution of the available Common Era RSL data at each site (Fig. 4, Supplementary Fig. 13). In northern New Jersey, the amplitude of local-scale sea-level change is <2 cm and it has the smallest contribution to RSL, with rates ranging from −0.1 to 0.1 mm yr$^{-1}$ throughout the Common Era (Table 1). At all sites, the amplitude of local-scale sea-level change is <16 cm and its contribution to RSL change ranges from −0.3 to 0.3 mm yr$^{-1}$, with persistent positive or negative contributions for several centuries at a time, and the greatest contributions in New York City and North Carolina.

Local-scale sediment compaction can contribute to RSL change, but the sequences from these sites have been decompacted (e.g., Supplementary Fig. 6), and furthermore, sediment compaction has been shown to have a minimal effect on RSL reconstructions from continuous sequences of high salt-marsh peat with small overburden[16,37]. Anthropogenic groundwater withdrawal can cause local-scale RSL differences across geographically proximal locations[24]. Coastal New Jersey has been shown to experience up to ~0.7 mm yr$^{-1}$ of subsidence due to groundwater withdrawal in the 20th century[24], which could explain some of the local contribution observed in northern and southern New Jersey in the last century. The effects of groundwater withdrawal-induced subsidence are likely highly localized, however, considering the NOAA-operated tide gauge at Sandy Hook, New Jersey (station number 8531680), located ~20 km from Cheesequake State Park (Fig. 1b, d), shows a rate of rise of 4.1 ± 0.1 mm yr$^{-1}$ from 1940–2000 CE compared to the proxy-based reconstruction at Cheesequake of 3.2 ± 0.8 mm yr$^{-1}$.

Tidal range changes through time could be a contributing factor to local differences in RSL because the foraminifera indicators used as a proxy to reconstruct RSL are linked to

modern tidal levels[38,39]. Therefore, if the local tidal range differed in the past, the RSL reconstructions using these tide-level indicators will not match the true RSL curve[40]. Changing bathymetric depths or coastline shapes from the effects of sedimentation can affect tidal ranges over centuries to millennia[41,42]. More recent tidal range changes, such as those observed at tide-gauge sites worldwide in the last century, due to natural or anthropogenic changes such as a loss of wetlands, dredging, or changes in sedimentation due to deforestation could also contribute to changes in the local component over the 20th century[43]. For example, in New York City, the local component underwent an abrupt shift during the last century. The six U.S. Atlantic coast sites analysed here were not corrected for tidal range changes, but they were reconstructed from depositional environments with small tidal ranges that have been shown to produce the most precise RSL reconstructions[44]; therefore, even a large percentage change in tidal range would have a minimal influence on the absolute elevation of foraminiferal indicators. Further, paleotidal modelling efforts on the U.S. Atlantic coast have demonstrated that past tidal range changes have minimal influence on rates of sea-level change, at least on basin scales; nevertheless, site-specific factors, such as the location of a site within a bay system, must be considered[41,42]. For example, in North Carolina, Kemp et al.[45] explores the influence of tidal range changes on RSL reconstructions from changes in paleogeography through the opening and closing of inlets[46] in the Outer Banks barrier islands. In the case of a doubling of tidal range through the opening of a barrier, the RSL reconstruction is altered on centimeter scales[45]. In New York City, Kemp et al.[15] used hydrodynamic modelling to examine the influence of tidal range changes as RSL rise likely increased the tidal range in Long Island Sound over the past 1500 years. A reconstruction adjusted for past tidal range change results in an average difference from the original reconstruction of 0.05 m (up to 0.11 m); however, the overall RSL trends remain the same as these changes are within the boundary of uncertainties[15]. Therefore, there is potential for tidal range change to influence reconstructed RSL rates, but the magnitude of RSL change is relatively small and the extent of its influence is site specific. In particular, the precise local-scale geomorphologic evolution at each individual site is needed to understand the full effects of potential tidal range changes driven by geomorphologic changes that could contribute to the local component of sea-level change.

In this study, we use the distinct spatial scales of processes driving sea-level change to distinguish their varying contributions through time. Along the U.S. Atlantic coast, Common Era RSL change is dominated by regional-scale linear processes due to the effects of GIA, with the fastest rates of rise of 1.6 ± 0.02 mm yr$^{-1}$ in southern New Jersey. Regional nonlinear changes

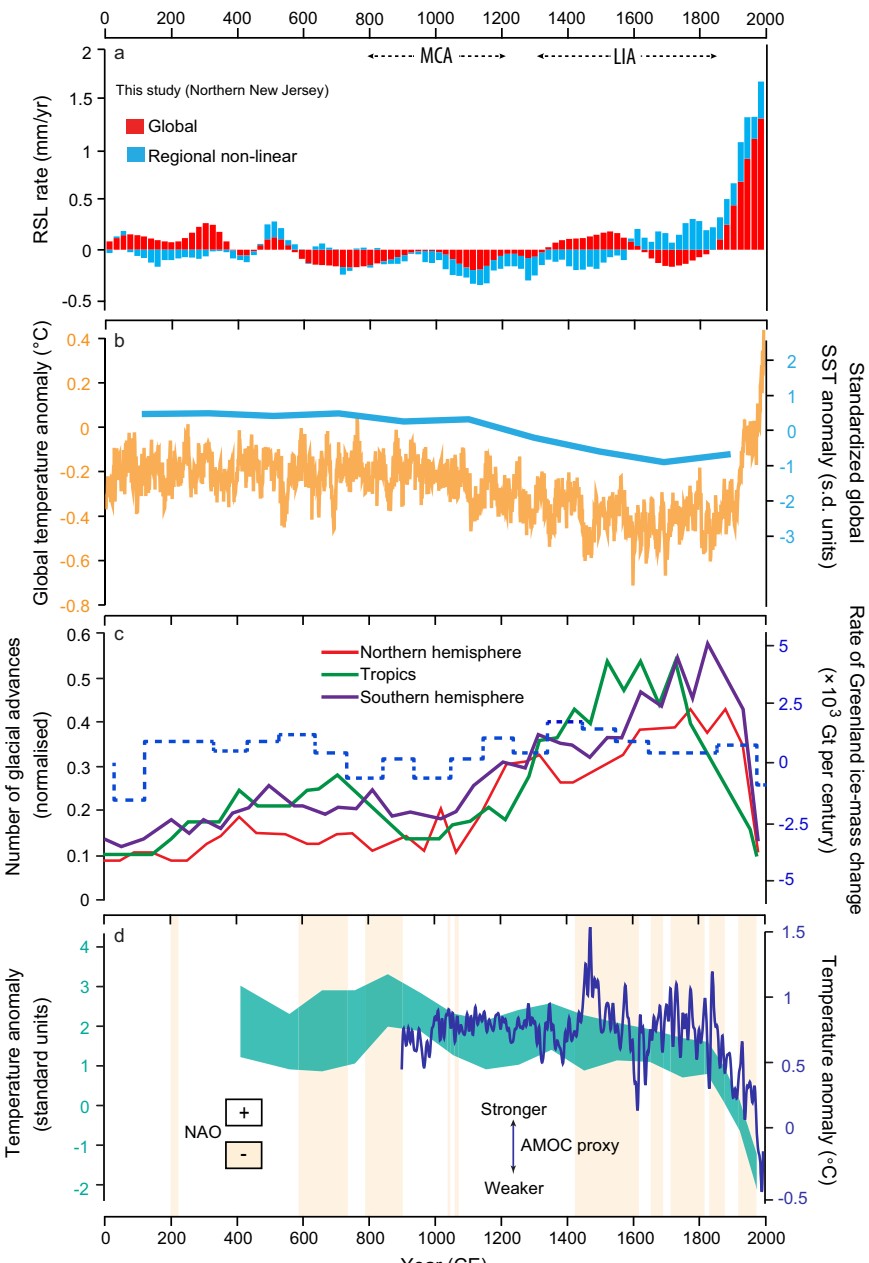

**Fig. 5 Sea-level budget compared with climate proxies over the Common Era. a** Global and regional nonlinear portion of the sea-level budget for northern New Jersey. RSL relative sea level. **b** PAGES2k global surface temperature anomaly from Neukom et al.[20] and standardized Ocean2k synthesis of median global sea surface temperature (SST) anomaly from McGregor et al.[68]. **c** Number of glacial advances by region from Solomina et al.[21] and rate of Greenland ice-mass change from Briner et al.[36]. **d** Atlantic Meridional Overturning Circulation (AMOC) proxy reconstructions from Rahmstorf et al.[69] (blue) and Thornalley et al.[32] (green) and positive and negative North Atlantic Oscillation (NAO) reconstruction adapted from Baker et al.[30]. Timing of Medieval Climate Anomaly (MCA) and Little Ice Age (LIA) from Neukom et al.[70].

from atmosphere/ocean dynamics and gravitational, rotational, and deformational effects of land-ice change are similar among the sites, contributing between −0.3 and 0.4 mm yr⁻¹ at each site through the Common Era. The consistency of this signal north of the North Carolina site argues that atmosphere/ocean dynamics are the dominant driver of this term. The local signal, likely primarily due to tidal range changes and anthropogenic groundwater withdrawal, vary spatially and temporally among sites between −0.3 and 0.3 mm yr⁻¹. The most significant feature of each sea-level budget is the recent redistribution of budget components due to the increasing contribution of the common global component, associated with the global-mean effects of thermosteric sea-level change and land-ice-mass loss, which

matches or surpasses the influence of GIA at each site in the 20th century, when the global signal reaches a rate of 1.3 ± 0.1 mm yr⁻¹. Based on these results on the U.S. Atlantic coast, this method could be applied to more sites globally to produce site-specific Common Era sea-level budgets, which could be used to resolve the spatially distinct processes in even greater detail.

## Methods

**Relative sea-level reconstructions**. The RSL reconstructions along the U.S. Atlantic coast (Fig. 1a) use salt-marsh foraminifera as a proxy because their modern distributions exhibit vertical zonation in relation to tidal levels[38,39]. Foraminiferal-based transfer functions utilize a modern foraminifera training set to quantify species assemblages' relationship with elevation, which is then applied to

sediment core fossil assemblages to produce continuous records of sea level at decadal and decimeter scale resolution[39].

We constructed a new Common Era RSL record from a salt-marsh site at Cheesequake State Park in northern New Jersey off of Raritan Bay, ~4 miles from the southern tip of Staten Island and ~23 miles from The Battery (Fig. 1, Supplementary Fig. 1). The foraminiferal-based transfer function for northern New Jersey uses a Bayesian approach that employs foraminifera, as well as bulk sediment $\delta^{13}C$ measurements as an additional constraint to reduce vertical uncertainty[15,47]. Stable carbon isotope geochemistry ($\delta^{13}C$) in bulk sediment represents the dominant vegetation type and can be used as a proxy for sea level because the transition between $C_3$- and $C_4$-dominated salt-marsh plant communities has been shown to occur at the mean higher high water (MHHW) tidal datum on the U.S. mid-Atlantic coast[48]. The Bayesian transfer function (BTF) was developed using a New Jersey modern training set of salt-marsh foraminifera and $\delta^{13}C$ in southern New Jersey from Kemp et al. (2013), in addition to 32 modern samples collected at Cheesequake State Park (Supplementary Figs. 2, 3). We measured bulk sediment $\delta^{13}C$ on modern and down-core samples at Bryn Mawr College using cavity ring-down laser spectroscopy using techniques following Balslev-Clausen et al.[49]. We calibrated the BTF using the combined New Jersey modern training set and evaluated its performance using cross-validation[47] (Supplementary Fig. 4). We formally accounted for temporal and spatial variability of modern foraminifera distributions in the BTF by including informative foraminifera variability priors for individual species using data from a monitoring study of modern foraminifera in southern New Jersey[50]. The BTF was applied to sediment core foraminifera and $\delta^{13}C$ data to provide paleomarsh elevation (PME) estimates with 95% credible intervals for each core sample (Supplementary Fig. 5). We calculated RSL by subtracting the PME estimates from the sample altitude. In addition, we used a geotechnical model[51,52] to correct the RSL record for post-depositional lowering through sediment compaction (Supplementary Fig. 6) that has been used previously to correct salt-marsh RSL reconstructions for compaction[16].

We reconstructed RSL using the transfer function estimates of PME in combination with a sediment core chronology. A sediment core chronology was constructed using Accelerator Mass Spectrometry radiocarbon ($^{14}C$) dating on identifiable plant macrofossils (stems and rhizomes) in the sediment core (Supplementary Table 1). A plateau in the radiocarbon calibration curve often results in radiocarbon dated material from the past ~300 years having multimodal age estimates and large uncertainties[53]. Therefore, to provide a chronology for the last several hundred years, we used changes in *Ambrosia* pollen abundances, regional-scale pollution markers (recognized in changes in down-core concentrations of lead, copper, cadmium, and nickel), the ratio of lead isotopes ($^{206}Pb$:$^{207}Pb$), and $^{137}Cs$ activity (Supplementary Fig. 7). Radiocarbon dates, pollen abundances, pollution markers, and $^{137}Cs$ activity were compiled using the Bchron package in R[54,55], which uses a Bayesian framework to produce an age-depth model and estimates ages with associated uncertainties for every 1-cm-thick interval in the core (Supplementary Fig. 8). The age estimates and uncertainties from Bchron were applied to all core samples with a reconstructed PME (Supplementary Fig. 9). The data for the northern New Jersey RSL record can be found in the Supplementary Materials.

**Spatiotemporal statistical model.** To estimate past RSL and rates of RSL change and their associated uncertainties, we used a spatiotemporal empirical hierarchical model[11,17] with a sea-level database comprising proxy sea-level records with high-resolution chronologies from 36 regions around the world, including the new northern New Jersey RSL record (Fig. 3a). The 2274 individual data points in the database use proxies such as foraminifera, diatoms, testate amoebae, coral micro-atolls, archaeological evidence, and sediment geochemistry. We have updated the database (Supplementary Data) from Kemp et al. (2018) to include 390 new RSL data points from northern New Jersey, USA (this study); Croatia[56]; French Polynesia[57]; Israel[58]; Quebec[59]; Connecticut[25]; Maine[25]; and Nova Scotia[25]. We compared the new northern New Jersey record with other published records along the U.S. Atlantic coast from southern New Jersey (Leeds Point and Cape May Courthouse[14]), New York City[15], Connecticut[60], and North Carolina[45] to examine regional variability in magnitudes and rates of past RSL change, as well as variability in regional Common Era sea-level budgets.

As in Kopp et al.[11] and Kemp et al.[16], decadal-average values from instrumental tide-gauge records in the Permanent Service for Mean Sea Level (PSMSL)[61] were included in the analysis, provided they were either (1) longer than 150 years, (2) within 5 degrees distance of a proxy site and longer than 70 years, or (3) the nearest tide gauge to a proxy site that is longer than 20 years[11,16]. We also include multicentury records from Amsterdam (1700–1925 CE)[62], Kronstadt (1773–1993 CE)[63], and Stockholm (1774–2000 CE)[64], as compiled by PSMSL. The input data also include the global-mean sea-level reconstruction of Hay et al.[18] from tide-gauge records for 1880–2010.

The model has (1) a process level that characterizes RSL over space and time and (2) a data level that links RSL observations (reconstructions) to the RSL process. Hyperparameters characterize prior expectations of dominant spatial and temporal scales of RSL variability, set through maximum-likelihood optimization.

At the process level, the RSL field $f(\boldsymbol{x}, t)$ is modelled as the sum of seven components[16]:

$$f(\boldsymbol{x}, t) = g_f(t) + g_s(t) + m(\boldsymbol{x})(t - t_0) + r_s(\boldsymbol{x}, t) + r_f(\boldsymbol{x}, t) + l_s(\boldsymbol{x}, t) + l_f(\boldsymbol{x}, t)$$

(1)

where $\boldsymbol{x}$ represent geographic location, $t$ represents time, and $t_0$ is a reference time point (2000 CE). The seven components include fast and slow common global (or globally uniform) terms ($g_f(t)$ and $g_s(t)$), a regional linear term ($m(\boldsymbol{x})(t - t_0)$), fast and slow regional nonlinear terms ($r_f(\boldsymbol{x}, t)$ and $r_s(\boldsymbol{x}, t)$), and fast and slow local terms ($l_f(\boldsymbol{x}, t)$ and $l_s(\boldsymbol{x}, t)$). The regional linear term uses predictions from the ICE5G–VM2– 90 Earth-ice model[65] as prior means. We also ran the model using predictions from the ICE6G–VM5a model[66], but the differences in regional linear rates were negligible (Supplementary Table 3).

The data level includes the RSL reconstructions with observations, $y_i$, where:

$$y_i = f(\boldsymbol{x}_i, t_i) + y_0(\boldsymbol{x}_i) + \varepsilon_i + w(\boldsymbol{x}_i, t_i)$$

(2)

$$t_i = \widehat{t_i} + \delta_i$$

(3)

where $f(\boldsymbol{x}_i, t_i)$ is the true RSL value at location $\boldsymbol{x}$ and time $t$, $y_0(\boldsymbol{x}_i)$ is a site-specific vertical datum correction to ensure that the RSL reconstructions are directly comparable to one another, $\varepsilon_i$ is the vertical uncertainty, which is treated as independent and normally distributed (with a standard deviation for each data point from the original publication), and $w(\boldsymbol{x}_i, t_i)$ is supplemental white noise. The true age of a RSL observation ($t_i$) is the mean estimate ($\widehat{t_i}$) and its error ($\delta_i$).

The hyperparameters characterize prior expectations of amplitudes and spatial and temporal scales of RSL variability (Supplementary Table 2). These amplitudes and scales are estimated using a maximum-likelihood optimization. The nonlinear terms were characterized by three spatial scales (global, regional, and local) and two temporal scales (fast and slow). These different spatial and temporal scales allow RSL to be decomposed into common global, regional temporally linear, regional nonlinear, and local components. As in Kopp et al.[11] and Kemp et al.[16], we apply a constraint on the model that mean global sea level over −100–100 CE is equal to mean global sea level over 1600–1800 CE because a constant global rate could also be interpreted as a regional linear trend (Supplementary Fig. 14). The decomposed components are used to produce Common Era sea-level budgets for northern New Jersey and five other sites along the U.S. Atlantic coast using 100-year average rates in 20-year timesteps to examine the evolving contribution of each component through time.

We also conduct several sensitivity tests on the model. We ran a sensitivity check to test the prior specifications by fixing the prior amplitudes for all nonlinear terms to be the same and then optimizing all of the hyperparameters; the objective is to maximize the log likelihood and in this case with new hyperparameter values, the posterior likelihood decreased (log likelihood changed from −10352 to −10438), and RSL trends remained largely consistent. Additionally, we assess the effect of initial values (within the upper to lower bounds) of each hyperparameter on the optimized maximum-likelihood values. We find that different initial values have very little influence on the optimized hyperparameters: differences in timescale estimates were <2 years, differences in length scale estimates were <0.03 degrees, and differences in prior SD were <0.3 cm. We also performed a "leave-one-site-out" cross-validation of the six study sites using the original optimized hyperparameters, individually removing the data from each site and predicting the RSL change at each site given the rest of the data. Overall the model is reasonably well calibrated with a coverage probability based on a "leave-one-site-out" validation of 88%. The mean error is −0.105 m, suggesting that the model tends to over predict on average and the mean absolute error is 0.165 m (Supplementary Table 4). Further, the rate predictions at the other five sites remained consistent and the uncertainties increased by <0.01 mm/yr (Supplementary Fig. 15). Therefore, the model results are generally robust to the removal of site data. Additionally, Supplementary Fig. 16 illustrates a comparison of RSL predictions for northern New Jersey using different model variations, including removing the northern New Jersey data and predicting RSL, using only the northern New Jersey data, and reoptimizing hyperparameters using only the northern New Jersey data.

## Data availability
Data related to this article can be found in the Supplementary Information and Supplementary Data file. Source data are provided with this paper.

## Code availability
Code for the spatiotemporal model results that are reported in the paper are available at https://github.com/bobkopp/CESL-STEHM-GP(10.5281/zenodo.4549924 for CESL-STEHM-GP)[67].

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

## Acknowledgements

J.S.W. thanks Kristen Joyse, Ane Garcia-Artola, and Margaret Christie for their assistance in the field to produce the northern New Jersey record and Andrew Kemp for discussion of interpretation of the RSL data. J.S.W. was funded by the David and Arleen McGlade Foundation and a Cushman Foundation for Foraminiferal Research Student Research Award. J.S.W. and R.E.K. were supported by the U.S. National Science Foundation (NSF) award OCE-1804999. R.E.K. and E.L.A. were supported by NSF grants OCE-1702587 and OCE-2002437. B.P.H. and T.S. are supported by the Singapore Ministry of Education Academic Research Fund MOE2019-T3-1-004, the National Research Foundation Singapore, and the Singapore Ministry of Education, under the Research Centers of Excellence initiative. N.C. is supported by the A4 project (Grant-Aid Agreement No. PBA/CC/18/01) carried out with the support of the Marine Institute under the Marine Research Programme funded by the Irish Government. D.C.B. receives support from the H.F. Alderfer Fund for Environmental Studies at Bryn Mawr College. We acknowledge PALSEA (Palaeo-Constraints on Sea-Level Rise), a working group of the International Union for Quaternary Sciences (INQUA) and Past Global Changes (PAGES), which in turn received support from the Swiss Academy of Sciences and the Chinese Academy of Sciences. This article is a contribution to HOLSEA (Geographic variability of Holocene sea level) and International Geoscience Program (IGCP) Project 639, "Sea-Level Changes from Minutes to Millennia". This work is Earth Observatory of Singapore contribution 354.

## Author contributions

J.S.W., R.E.K., and B.P.H. designed the research approach. J.S.W. led the research and wrote the first draft of the paper. J.S.W., R.E.K., T.A.S., N.C., N.S.K., D.C.B., E.L.A., M.B., J.L.C., D.R.C., and B.P.H. contributed to producing the new relative sea-level reconstruction in northern New Jersey and to the writing of the manuscript.

## Competing interests

The authors declare no competing interests.

## Additional information

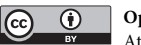

