## [Peer Review File · Nature Communications]

REVIEWER COMMENTS

Reviewer #1 (Remarks to the Author):

The paper analyses the different contributions to change in sea level from years 0 to 2000 CE, focusing on U.S. Atlantic coast. The analysis is based on the decomposition of RSL records for the identification of linear/non-linear and global/regional components. The work proposes an interesting approach that helps for a better comprehension of the different mechanisms acting on sea level change. The time span chosen for the analysis is adequate to investigate the changes in relative importance of the different factors acting on sea level from pre-industrial to industrial era. My only suggestion is to reinforce the discussion about the causes of identified components, corroborating with data the statements done. The different contributions to sea level are well studied (see, for example Stammer, Detlef, et al. "Causes for contemporary regional sea level changes", but a lot of recent studies on this topic are available). The added value of this work should be to compare modelled or observed trend for the different causes, at global or regional level (i.e. GIA, current ice melting, steric effects, sedimentation, seismic and co-seismic deformation, etc) with the trend for the components extracted from RSL records. This could also help to further motivate the conclusions, that appears a bit speculative.

Minor comments:

L54. Common era (last 2000 years): to be precise, this not the exact definition of Common Era and does not correspond to the analysis done in the paper (last 2000 years are from 20 CE to 2020 CE..., the paper is referred to 0-2000 CE).

LI 84-91: trend at different location shows a north-south gradient, in line with the post glacial rebound signals, with the exception of North Carolina sites. Do the Authors have some speculation about the reason?

LI 102-105: different studies on past sea-level have identified multidecadal periodic oscillations. Has the authors considered the possible influence of these on the different trends identified from 1800 CE? (see for example Chambers, D. P., Merrifield, M. A., & Nerem, R. S. (2012) *Geophysical Research Letters*, 39(18).

LL 113-120: thermal expansion of oceans has a strong regional dimension; the changes in mass and volume of oceans from the melting of glaciers and ice caps implies regional changes in sea level (for the well known mechanisms linked to mass redistribution). Please, discuss what you refer for the global fraction of these causes.

LI 132-134: the natural compaction rate is consistent with the trend for the extracted component, considering the GIA rate?

LI 163-171: Agree, only two marginal considerations. The behaviour of Greenland ice sheet, even if not perfectly constrained, is well known, and studies confirm its non-linear dynamic, even if with a neat tendency to melting in the industrial era. The difficulty in detecting the Greenland signal in the regional non-linear component is probably due to its low non-linearity (compared to the other causes identified, i.e. steric effects and change in ocean circulation) more than to the magnitude of the effects of ice melting on sea level change.

LI192-202: it is not clear here how authors have included considerations about tidal range (and its potential effect in reconstruction) in their analysis.

LI 491: the ICE-5G model is now commonly replaced by the ICE-6G model (Peltier, W. R., et al. "Postglacial rebound and current ice loss estimates from space geodesy: the new ICE-6G (VM5a)

global model." AGUFM 2012 (2012): G23C-02.) Furthermore, are available results for the ICE-7G version for North America (<https://www.atmosph.physics.utoronto.ca/~peltier/data.php>)

Reviewer #2 (Remarks to the Author):

In this paper, the authors use a spatiotemporal hierarchical model to study rates of relative sea-level (RSL) rise along the U.S. Atlantic coast over the past 2000 years based on proxy and instrumental sea-level records, including a new proxy record in northern New Jersey. The rates of rise are decomposed into contributions from various processes according to their different characteristic spatial and time scales. The main findings are a dominant role of regional temporally linear processes, primarily glacial isostatic adjustment, through the Common Era, and the increasing importance of the global signal since 1800, which by 1950 explains up to 50% of the RSL change.

I enjoyed reading this well-written paper and, being someone who works primarily with instrumental records and modeled data, I find it fascinating that RSL can be reconstructed with such a high precision using proxies. Understanding how and why sea level is changing on regional and local scales is a major intellectual challenge in the climate sciences, but it also has enormous practical importance, and so the results of this paper are likely to appeal to a broad audience, including scientists and the general public. The authors use well-established and previously published methods to arrive at their results. I support publication of this paper in Nature Communications. However, I have a number of concerns and questions of clarification, detailed below, that should be addressed prior to publication.

Specific comments:

1. The explanatory analysis of the sea-level budgets in terms of possible causes is fairly speculative and I think that more fleshing out is needed to persuade the reader that the mechanisms being proposed have merit. I do recognize the difficulty of attributing the rates to specific causes given that there is not an exclusive one-to-one correspondence between the spatiotemporal processes in the statistical model and the actual mechanistic processes, but I feel that some of the statements regarding causation are very important and so need to be more convincing. This could be achieved simply by extending the discussion to clarify whether the estimated changes are consistent with what would be expected from the proposed drivers. For example, the authors suggest that NAO changes could explain part of the regional non-linear contribution, so why not to plot the NAO reconstruction from Ortega et al. (2015), which is publicly available, on top of the sea-level budgets to see if the sea-level and NAO are consistent with what we would expect? The authors also vaguely state that the AMOC can cause sea-level changes on the U.S. Atlantic coast, but they could be more specific and mention that the AMOC weakening over the industrial period (Dima and Lohmann, 2010; Cheng et al., 2013; Rahmstorf et al., 2015) could be responsible for the positive non-linear contribution over the same period. Are these changes consistent with the expectation that a 2 cm rise in sea level corresponds to a 1Sv drop in the AMOC? These are just a few examples

2. All the results presented here are entirely based on a spatiotemporal hierarchical model, yet there seems to be little effort towards validating the model, which I think is crucial to ensuring reliable inferences. At the very least the authors should check that the model is consistent with the data: do data simulated from the posterior predictive distribution look similar to the observations? I would also suggest that they conduct a sensitivity analysis. For example, in the context of Gaussian processes, changing the hyperparameters even by a small amount will lead to a set of functions that has no overlap with the original space of functions. To put it simply, hyperparameters with slightly different values can yield substantially different results. This is not an issue when using a full Bayesian approach since it considers the entire space of functions but it can be problematic in empirical Bayes, especially because estimates of the hyperparameters from maximum likelihood tend to be sensitive to the starting values. Hence the question is, how sensitive are estimates of the hyperparameters to the starting initial values? And how sensitive are your posterior estimates to the value of the

hyperparameters? Do your posterior estimates remain consistent if you remove one of the records, for example that in northern New Jersey?

3. I understand from Kopp et al. (2016) that the vertical errors of the RSL observations, E_i , are treated as independent and normally distributed. If this were the case, I would expect to see a much larger spread of the observations around the mean in Figure 1d and Supplementary Figure 9. In other words, the standard deviation of the observations appears to be much smaller than the error bars. Am I misinterpreting what these error bars mean? Could you please plot $E_i = y_i - (f + w + y_0)$ (Equation 2) to see what it looks like? On a related comment, can E_i really be treated as independent considering that each data point comes from the same sampled core?

Figure 1d and Supplementary Fig. 9, could you please clarify in the Figure caption whether the uncertainties denote 1 or 2 sigma?

Lines 84-91. There seems to be a latitudinal gradient with rates of rise decreasing from south to north, but North Carolina does not fit this pattern. Any idea why?

Lines 161-163. Where exactly did the cooling take place? Several studies have suggested that cooling of the subpolar North Atlantic on long timescales is associated with an AMOC slowdown, which should lead to increased rates of sea-level rise along the U.S. Atlantic coast. Can this be reconciled with the mechanism being proposed here?

Lines 197-199. Some studies have found changes in the tidal range of more than 25 cm over the 20th Century (e.g., Mawdsley et al., 2015), including at some tide gauge stations along the U.S. Atlantic coast. Could these play a role here?

Lines 465-468. Closing parenthesis missing.

Lines 479-482. Strictly speaking this is a two-level model since no prior distributions are ascribed to the parameters.

Lines 484-491. To make the processes more mechanistically interpretable, here I would suggest that you say what physical process each of the terms in Equation (1) might represent. Also, please clarify how the processes are modeled (Gaussian processes?).

Line 488. I understand that the global terms represent a signal that is common to all data sites. However, all sources of global mean sea level give rise to spatially non-uniform sea-level changes. Hence it is not entirely clear to me what exactly the global and regional terms represent, since I imagine that the regional term will capture signals contributing to the global average such as the fingerprints of land-ice melting or non-uniform ocean warming. How does the model decide what goes to 'global' and what goes to 'regional'? Could you please clarify.

Lines 497-500. Please state whether E_i is treated as independent and normally distributed.

Lines 502-504. The length scale for the local processes is 0.04 degrees, which seems to be smaller than the minimum distance between data sites. Are you sure there is information in the data to say anything about these very short length scales?

Are the authors going to make the code publicly available? I think it would be a very valuable contribution to the scientific community.

References

Cheng W., Chiang J.C., Zhang D. (2013). Atlantic meridional overturning circulation (AMOC) in CMIP5 models: RCP and historical simulations. *J Clim* 26(18):7187–7197.
Dima M., Lohmann G. (2010). Evidence for two distinct modes of large-scale ocean circulation

changes over the last century. *J Clim* 23(1):5–16.

Mawdsley, R. J., Haigh, I. D., & Wells, N. C. (2015). Global secular changes in different tidal high water, low water and range levels. *Earth's Future*, 3(2), 66–81.

Rahmstorf S., Box J.E., Feulner G., Mann M.E., Robinson A., Rutherford S., Schaffernicht E.J. (2015). Exceptional twentieth-century slowdown in Atlantic Ocean overturning circulation. *Nat Clim Change* 5:475–480.

REVIEWER COMMENTS

Reviewer #1 (Remarks to the Author):

The paper analyses the different contributions to change in sea level from years 0 to 2000 CE, focusing on U.S. Atlantic coast. The analysis is based on the decomposition of RSL records for the identification of linear/non-linear and global/regional components. The work proposes an interesting approach that helps for a better comprehension of the different mechanisms acting on sea level change. The time span chosen for the analysis is adequate to investigate the changes in relative importance of the different factors acting on sea level from pre-industrial to industrial era.

My only suggestion is to reinforce the discussion about the causes of identified components, corroborating with data the statements done. The different contributions to sea level are well studied (see, for example Stammer, Detlef, et al. "Causes for contemporary regional sea level changes", but a lot of recent studies on this topic are available). The added value of this work should be to compare modelled or observed trend for the different causes, at global or regional level (i.e. GIA, current ice melting, steric effects, sedimentation, seismic and co-seismic deformation, etc) with the trend for the components extracted from RSL records. This could also help to further motivate the conclusions, that appears a bit speculative.

We expanded all discussion of the sea-level budgets to strengthen the connection between model components and potential contributing physical processes. Specifically, in Ln 137-141, we expanded the discussion of the processes contributing to the global component. In Ln 189-227, we expanded the discussion of processes contributing to the regional non-linear component to describe how the sea-level changes could be caused by climate in the North Atlantic and changes in NAO and AMOC or ice mass changes. Additionally, we expanded the discussion of processes contributing to the local component in Ln 261-285 to more completely describe how tidal range and geomorphologic changes could influence the local sea-level changes.

We also produced an additional figure, Figure 5, which compares the budget with various climate proxies (e.g. temp, NAO, AMOC), to illustrate the expanded discussion of processes.

Minor comments:

L54. Common era (last 2000 years): to be precise, this not the exact definition of Common Era and does not correspond to the analysis done in the paper (last 2000 years are from 20 CE to 2020 CE..., the paper is referred to 0-2000 CE).

We changed this in Ln 63 to "Common Era (0-2000 CE)" so that the timeframe corresponds to the analysis done in the paper.

L1 84-91: trend at different location shows a north-south gradient, in line with the post glacial rebound signals, with the exception of North Carolina sites. Do the Authors have some speculation about the reason?

We added in Ln 104-106 describing the spatial gradient of RSL rates as due to each location's position relative to the time-evolving position of the former Laurentide Ice Sheet.

L1 102-105: different studies on past sea-level have identified multidecadal periodic oscillations. Has the authors considered the possible influence of these on the different trends identified from 1800 CE? (see for example Chambers, D. P., Merrifield, M. A., & Nerem, R. S. (2012) *Geophysical Research Letters*, 39(18).

In this study, we were only focused on 100-year average trends, and while global sea level did vary on multidecadal to multicentennial timescales, we did not identify any consistent multidecadal periodicity in the magnitude or rates of global sea-level change over the Common Era and found only increasing global sea level since 1800 CE. We added in Ln 130-131 to state this. Additionally, we added in Ln 168-172, that the total nonlinear signal for each site also varies on multidecadal to multicentennial timescales, but again, we did not observe any consistent periodicity.

LL 113-120: thermal expansion of oceans has a strong regional dimension; the changes in mass and volume of oceans from the melting of glaciers and ice caps implies regional changes in sea level (for the well known mechanisms linked to mass redistribution). Please, discuss what you refer for the global fraction of these causes.

The model distinguishes between sea-level changes that are common across all sites and those that differ among regions or on smaller spatial scales. We use the term 'global' to refer only to the trend components that are common across all locations in the sea-level database. Sea-level changes that are unique to a particular region, such as a specific fingerprint from ice melt, are assigned to regional or local scales. We added in Ln 129-130 and Ln 144 to clarify this.

L1 132-134: the natural compaction rate is consistent with the trend for the extracted component, considering the GIA rate?

We state in Ln 164 that in coastal New Jersey, natural compaction has an average 20th century rate of 0.16 mm yr⁻¹, while we state in Ln 248 that the extracted groundwater withdrawal component may contribute up to ~0.7 mm yr⁻¹ of subsidence in the 20th century. These compare to the much larger linear rate of 1.5-1.6 mm yr⁻¹ due to GIA as stated in Ln 155.

L1 163-171: Agree, only two marginal considerations. The behaviour of Greenland ice sheet, even if not perfectly constrained, is well known, and studies confirm its non-linear dynamic, even if with a neat tendency to melting in the industrial era. The difficulty in detecting the Greenland signal in the regional non-linear component is probably due to its low non-linearity (compared to the other causes identified, i.e. steric effects and change in ocean circulation) more than to the magnitude of the effects of ice melting on sea level change.

We added text in Ln 215-227 to make these considerations more clear in the discussion of the Greenland Ice Sheet and also noted that the behaviour of the Antarctic Ice Sheet is even less constrained over this time period.

L1192-202: it is not clear here how authors have included considerations about tidal range (and its potential effect in reconstruction) in their analysis.

We expanded the discussion of the influence of tidal range in Ln 261-285 to describe the importance of site-specific analysis of tidal range change. We include specific examples from

New York and North Carolina to illustrate the potential impact of tidal range change on the proxy RSL reconstructions.

L1 491: the ICE-5G model is now commonly replaced by the ICE-6G model (Peltier, W. R., et al. "Postglacial rebound and current ice loss estimates from space geodesy: the new ICE-6G (VM5a) global model." AGUFM 2012 (2012): G23C-02.) Furthermore, are available results for the ICE-7G version for North America (<https://www.atmosphysics.utoronto.ca/~peltier/data.php>)

We ran the model using ICE-6G predictions and found that the differences in Common Era regional linear rates from the spatiotemporal model were negligible. We added Ln 390-392 to state this and Supplementary Table 3 to compare the results using ICE-5G and ICE-6G. The GIA predicted present-day sea-level change rates are not yet available for ICE-7G.

Reviewer #2 (Remarks to the Author):

In this paper, the authors use a spatiotemporal hierarchical model to study rates of relative sea-level (RSL) rise along the U.S. Atlantic coast over the past 2000 years based on proxy and instrumental sea-level records, including a new proxy record in northern New Jersey. The rates of rise are decomposed into contributions from various processes according to their different characteristic spatial and time scales. The main findings are a dominant role of regional temporally linear processes, primarily glacial isostatic adjustment, through the Common Era, and the increasing importance of the global signal since 1800, which by 1950 explains up to 50% of the RSL change.

I enjoyed reading this well-written paper and, being someone who works primarily with instrumental records and modeled data, I find it fascinating that RSL can be reconstructed with such a high precision using proxies. Understanding how and why sea level is changing on regional and local scales is a major intellectual challenge in the climate sciences, but it also has enormous practical importance, and so the results of this paper are likely to appeal to a broad audience, including scientists and the general public. The authors use well-established and previously published methods to arrive at their results. I support publication of this paper in Nature Communications. However, I have a number of concerns and questions of clarification, detailed below, that should be addressed prior to publication.

Specific comments:

1. The explanatory analysis of the sea-level budgets in terms of possible causes is fairly speculative and I think that more fleshing out is needed to persuade the reader that the mechanisms being proposed have merit. I do recognize the difficulty of attributing the rates to specific causes given that there is not an exclusive one-to-one correspondence between the spatiotemporal processes in the statistical model and the actual mechanistic processes, but I feel that some of the statements regarding causation are very important and so need to be more convincing. This could be achieved simply by extending the discussion to clarify whether the estimated changes are consistent with what would be expected from the proposed drivers. For example, the authors suggest that NAO changes could explain part of the regional non-linear contribution, so why not to plot the NAO reconstruction from Ortega et al. (2015), which is

publicly available, on top of the sea-level budgets to see if the sea-level and NAO are consistent with what we would expect? The authors also vaguely state that the AMOC can cause sea-level changes on the U.S. Atlantic coast, but they could be more specific and mention that the AMOC weakening over the industrial period (Dima and Lohmann, 2010; Cheng et al., 2013; Rahmstorf et al., 2015) could be responsible for the positive non-linear contribution over the same period. Are these changes consistent with the expectation that a 2 cm rise in sea level corresponds to a 1Sv drop in the AMOC? These are just a few examples

We extended all sections describing the sea-level budgets to strengthen the discussion of the processes contributing to the modelled rates of each budget component. Specifically, in Ln 137-141, we expanded the discussion of the processes contributing to the global component. In Ln 189-227, we expanded the discussion of processes contributing to the regional non-linear component to describe how the sea-level changes could be caused by climate in the North Atlantic and changes in NAO and AMOC or ice mass changes. Additionally, we expanded the discussion of processes contributing to the local component in Ln 261-285 to more completely describe how tidal range and geomorphologic changes could influence the local sea-level changes.

We also produced an additional figure, Figure 5, which compares the budget with various climate proxies (e.g. temp, NAO, AMOC), to illustrate the expanded discussion of processes.

2. All the results presented here are entirely based on a spatiotemporal hierarchical model, yet there seems to be little effort towards validating the model, which I think is crucial to ensuring reliable inferences. At the very least the authors should check that the model is consistent with the data: do data simulated from the posterior predictive distribution look similar to the observations? I would also suggest that they conduct a sensitivity analysis. For example, in the context of Gaussian processes, changing the hyperparameters even by a small amount will lead to a set of functions that has no overlap with the original space of functions. To put it simply, hyperparameters with slightly different values can yield substantially different results. This is not an issue when using a full Bayesian approach since it considers the entire space of functions but it can be problematic in empirical Bayes, especially because estimates of the hyperparameters from maximum likelihood tend to be sensitive to the starting values. Hence the question is, how sensitive are estimates of the hyperparameters to the starting initial values? And how sensitive are your posterior estimates to the value of the hyperparameters? Do your posterior estimates remain consistent if you remove one of the records, for example that in northern New Jersey?

We ran a sensitivity check to test the prior specifications by fixing the prior amplitudes for all nonlinear terms to be the same and then optimizing all of the hyperparameters; the objective is to maximize the log likelihood and in this case with new hyperparameter values, the posterior likelihood decreased (log-likelihood changed from -10352 to -10438), and RSL trends remained largely consistent (Ln 416-422).

We conducted additional tests with the hyperparameters by changing the starting initial values in the optimization. We selected different initial values (within the upper to lower bounds) for each hyperparameter and re-optimized all of the hyperparameters to see how they change. We find that different initial values have very little influence on the optimized hyperparameters: differences in timescale estimates were <2 years, differences in length scale estimates were <0.03 degrees, and differences in prior SD were <0.3 cm. We added Ln 423-427 stating these results.

We also conducted further sensitivity tests by removing individual records. We performed a “leave-one-site-out” cross-validation of the six study sites using the original optimized hyperparameters, individually removing the data from each site and predicting the RSL change at each site given the rest of the data. Overall the model is reasonably well calibrated with a coverage probability based on a “leave-one-site-out” validation of 88%. The mean error is -0.105 m, suggesting that the model tends to over predict on average and the mean absolute error is 0.165 m. Further, the rate predictions at the other five sites remained consistent and the uncertainties increased by <0.01 mm/yr. Therefore, the model results are generally robust to the removal of site data. We added Ln 427-435 stating this and we produced an additional figure, Supplementary Figure 15, and additional table, Supplementary Table 4, showing these results.

3. I understand from Kopp et al. (2016) that the vertical errors of the RSL observations, E_i , are treated as independent and normally distributed. If this were the case, I would expect to see a much larger spread of the observations around the mean in Figure 1d and Supplementary Figure 9. In other words, the standard deviation of the observations appears to be much smaller than the error bars. Am I misinterpreting what these error bars mean? Could you please plot $E_i = y_i - (f + w + y_0)$ (Equation 2) to see what it looks like? On a related comment, can E_i really be treated as independent considering that each data point comes from the same sampled core?

In Figure 1d, the spatiotemporal model errors are much smaller than the errors from the RSL reconstruction itself because the model predictions are constrained by all of the other data in the database as well and not just the one site. Additionally, a robust trend can still be determined by the model even with noisy observations with larger uncertainties.

Below is a plot of $E_i = y_i - (f + w + y_0)$ (Equation 2) for the northern New Jersey site

Technically, E_i is not entirely independent because we have to assume that sea level has been monotonically rising at the site over time and we assume the law of superposition for the sediment sequences, so the single core used is one time series with the oldest sequences at the base and the youngest at the surface.

Figure 1d and Supplementary Fig. 9, could you please clarify in the Figure caption whether the uncertainties denote 1 or 2 sigma?

We clarified in the figure captions for Figure 1d and Supplementary Figure 9 that the spatiotemporal model predictions denote 1 sigma, the vertical RSL denotes 1 sigma, and the

chronological denotes 2 sigma.

Lines 84-91. There seems to be a latitudinal gradient with rates of rise decreasing from south to north, but North Carolina does not fit this pattern. Any idea why?

We added in Ln 104-106 describing the spatial gradient of RSL rates as due to each location's position relative to the time-evolving position of the former Laurentide Ice Sheet.

Lines 161-163. Where exactly did the cooling take place? Several studies have suggested that cooling of the subpolar North Atlantic on long timescales is associated with an AMOC slowdown, which should lead to increased rates of sea-level rise along the U.S. Atlantic coast. Can this be reconciled with the mechanism being proposed here?

The cooling took place broadly in the northern hemisphere and North Atlantic region. The effect on regional sea-level would depend on the source of the cooling. This broader cooling due to insolation forcing could drive a negative sea-level trend as suggested, while a slowdown of AMOC then causes secondary cooling in the high-latitude North Atlantic. Meanwhile, regional sea-level effects due to changes in ocean circulation could offset the negative influence on regional sea-level trends from the initial cooling. In this case, when multiple processes are probably at play and may have opposing influences on regional sea-level trends, it is difficult to distinguish and quantify the relative magnitude of each individual process using this sea-level budget approach. We expand on these points in Ln 189-194 and Ln 227-230.

Lines 197-199. Some studies have found changes in the tidal range of more than 25 cm over the 20th Century (e.g., Mawdsley et al., 2015), including at some tide gauge stations along the U.S. Atlantic coast. Could these play a role here?

These tidal range changes could be affecting the RSL reconstructions and therefore contributing to the local component over the 20th century. We added in Ln 261-266 the potential contribution of 20th century tidal range changes to the local component and we expanded on the overall influence of tidal range changes in Ln 261-285.

Lines 465-468. Closing parenthesis missing.

We added closing parenthesis in Ln 368.

Lines 479-482. Strictly speaking this is a two-level model since no prior distributions are ascribed to the parameters.

We updated the language in Ln 379-382 to describe the model as two-level.

Lines 484-491. To make the processes more mechanistically interpretable, here I would suggest that you say what physical process each of the terms in Equation (1) might represent. Also, please clarify how the processes are modeled (Gaussian processes?).

The spatiotemporal model only determines the spatial and temporal scales and trends within the RSL reconstructions. The results and discussion of the paper are to explain what physical processes each of the terms in Equation (1) might represent. Our assignment of potential physical processes to each of the terms represents an interpretation of the model result.

Therefore, we disagree with reviewer 2's suggestion to include physical processes for Equation (1) terms in the methods section describing the model.

The priors for each term in the model are Gaussian processes.

Line 488. I understand that the global terms represent a signal that is common to all data sites. However, all sources of global mean sea level give rise to spatially non-uniform sea-level changes. Hence it is not entirely clear to me what exactly the global and regional terms represent, since I imagine that the regional term will capture signals contributing to the global average such as the fingerprints of land-ice melting or non-uniform ocean warming. How does the model decide what goes to 'global' and what goes to 'regional'? Could you please clarify.

The model distinguishes between sea-level changes that are common across all sites and those that differ among regions or on smaller spatial scales. We use the term 'global' to refer only to the trend components that are common across all locations in the sea-level database. Sea-level changes that are unique to a particular region, such as a specific fingerprint from ice melt, are assigned to regional or local scales. We added in Ln 129-130 and Ln 144 to clarify this.

Lines 497-500. Please state whether E_i is treated as independent and normally distributed.

We added in Ln 400-401 that E_i is treated as independent and normally distributed.

Lines 502-504. The length scale for the local processes is 0.04 degrees, which seems to be smaller than the minimum distance between data sites. Are you sure there is information in the data to say anything about these very short length scales?

The length scale for local processes is set through maximum-likelihood optimization and it is smaller than the minimum distance between sites. In this way, the local component includes trends that are unique to an individual site. Common trends among sites on global and regional scales have been identified in the global and regional components of the model, so the local-scale processes are the residual trends that are only found at an individual site. We added in Ln 233-234 to clarify this. The local-scale interpretation is limited by the extent and resolution of the available data at each site as stated in Ln 235-236.

Are the authors going to make the code publicly available? I think it would be a very valuable contribution to the scientific community.

We added a 'Code availability' section stating where the code is publicly available.

We also added a 'Data availability' section where data related to the article can be found.

References

- Cheng W., Chiang J.C., Zhang D. (2013). Atlantic meridional overturning circulation (AMOC) in CMIP5 models: RCP and historical simulations. *J Clim* 26(18):7187–7197.
- Dima M., Lohmann G. (2010). Evidence for two distinct modes of large-scale ocean circulation changes over the last century. *J Clim* 23(1):5–16.
- Mawdsley, R. J., Haigh, I. D., & Wells, N. C. (2015). Global secular changes in different tidal high water, low water and range levels. *Earth's Future*, 3(2), 66–81.
- Rahmstorf S., Box J.E., Feulner G., Mann M.E., Robinson A., Rutherford S., Schaffernicht E.J.

(2015). Exceptional twentieth-century slowdown in Atlantic Ocean overturning circulation. *Nat Clim Change* 5:475–480.

REVIEWERS' COMMENTS

Reviewer #1 (Remarks to the Author):

[No additional comments--supports publication]

Reviewer #2 (Remarks to the Author):

I am generally happy with the revisions and would like to thank the authors for the efforts put into revising the paper, and particularly for responding to my questions, extending the discussion on processes, and adding the new Figure 5. I only have two further comments, both related to concerns I raised in my first review.

1. In my first review, I mentioned that the scatter of the RSL observations around the mean in Figure 1d and Supplementary Figure 9 is much smaller than what would be expected from the 1-sigma denoted by the uncertainty boxes. The answer given by the authors in the response letter does not address this issue and so I will expand on my remarks and try to make my point clearer. As I understand it, the error E_i in Equation (2) is assumed to be normally distributed with a standard deviation set equal to half the height of the boxes in Figure 1d, which is about 35 cm. However, the residuals from the model (i.e., $y_i - (f + w + y_0)$) appear to show deviations that are much smaller than 35 cm, and hence seem to be inconsistent with the assumed noise. This suggests that the normal observational model is inconsistent with the true data generating process. In a consistent model, the standard deviation of the time series of residuals ($y_i - (f + w + y_0)$) should be roughly similar to the assumed standard deviation for E_i (i.e., ~ 35 cm). And just to clarify, the small residuals cannot be explained through data pooling because E_i is treated as independent across space. These inconsistencies can have important inferential implications and make it difficult to say much about how meaningful the predictions are. Hence, I think that further explanation is needed here.

2. The authors refer to the terms $g_f(t)$ and $g_s(t)$ as the global components of RSL. However, these terms represent uniform changes in sea level, which are not equivalent to GMSL changes. The two main causes of GMSL changes are land-mass changes and ocean heat uptake and both give rise to non-uniform patterns of sea-level rise. For this reason, I think that the terminology used here is misleading and it would be more appropriate to call these terms "uniform components" of RSL. Related to this, in lines 118-120 the authors write "Our global sea-level estimate is consistent with other GMSL budgets for the 20th century". Is your "global sea-level estimate" given solely by the term $g_s(t)$? If yes, I wouldn't expect such estimate to be consistent with other GMSL since the former is missing important contributions to the global average associated with non-uniform changes. Could you please clarify?

Line 42. To be precise, the first Argo floats were deployed in 1999 and it was not until 2007 that the program achieved a reasonably good spatial coverage with 3000 floats.

REVIEWERS' COMMENTS

Reviewer #1 (Remarks to the Author):

[No additional comments--supports publication]

Reviewer #2 (Remarks to the Author):

I am generally happy with the revisions and would like to thank the authors for the efforts put into revising the paper, and particularly for responding to my questions, extending the discussion on processes, and adding the new Figure 5. I only have two further comments, both related to concerns I raised in my first review.

1. In my first review, I mentioned that the scatter of the RSL observations around the mean in Figure 1d and Supplementary Figure 9 is much smaller than what would be expected from the 1-sigma denoted by the uncertainty boxes. The answer given by the authors in the response letter does not address this issue and so I will expand on my remarks and try to make my point clearer. As I understand it, the error E_i in Equation (2) is assumed to be normally distributed with a standard deviation set equal to half the height of the boxes in Figure 1d, which is about 35 cm. However, the residuals from the model (i.e., $y_i - (f + w + y_0)$) appear to show deviations that are much smaller than 35 cm, and hence seem to be inconsistent with the assumed noise. This suggests that the normal observational model is inconsistent with the true data generating process. In a consistent model, the standard deviation of the time series of residuals ($y_i - (f + w + y_0)$) should be roughly similar to the assumed standard deviation for E_i (i.e., ~ 35 cm). And just to clarify, the small residuals cannot be explained through data pooling because E_i is treated as independent across space. These inconsistencies can have important inferential implications and make it difficult to say much about how meaningful the predictions are. Hence, I think that further explanation is needed here.

The fact that the standard deviation of the model can be smaller than the uncertainty of the individual observations can be shown by doing a simple fit of a linear model to the northern New Jersey data. As seen in the below figure, the uncertainties on the linear predictions (red lines) are substantially narrower than on the observations (gray lines). This is because we are trying to predict the latent variable (sea level), not the observations of sea level, which would have an additional error term. If we were examining model predictions of the observations of sea level, we would indeed expect the prediction uncertainties to be of comparable scale to those of the observations.

In looking specifically at the residuals, they do exhibit a temporal correlation for northern New Jersey (below, “All data included” shows these residuals when using the full database), which is a fair point. However, if we run a similar model with only the northern New Jersey data and re-optimize the hyperparameters, we no longer see a substantial temporal correlation. In the analysis in the paper, we are looking at the overall profile of the data across all sites in the database (not just northern New Jersey), which leads to the hyperparameters that we are using, and therefore to the classification of some of what one might consider as signal if looking at northern New Jersey in isolation as instead noise. In other words, our model is learning – from the entirety of the data set – that some of the apparent signal in the northern New Jersey record, viewed in isolation, is most probably idiosyncratic noise best attributable to observation error.

The influence on the northern New Jersey prediction can be further seen in the figure below with different model variations to produce Figure 1d. (a) “All data included” is using the entire database with the model as is done in the paper. (b) “Predicted Northern New Jersey” is removing the northern New Jersey data from the database, but keeping the rest of the data, and then predicting RSL at the northern New Jersey location (the proxy data is only shown on this figure as a reference). Here, the model is predicting northern New Jersey RSL and its uncertainties based on the other data in the database. The curve here is effectively the model’s prior for northern New Jersey before introducing our new record, and emphasizes that the rest of the database is quite informative with respect to this prior. (c) “Only Northern New Jersey” is

only using the data from northern New Jersey and removing the rest of the data in the database. In this case, we see slightly larger uncertainties in the predictions of RSL and a greater emphasis on shorter wavelength variability that is not supported by the corpus of sites in the region (and thus not seen in the previous case). (d) Finally, we ran the model as “Only Northern New Jersey” again, but this time with re-optimized hyperparameters, further emphasizing some centennial-scale variability that the more complete analysis smooths over. We added the below figure as Supplementary Figure 16 and added text in the methods in Ln 410-414.

2. The authors refer to the terms $g_f(t)$ and $g_s(t)$ as the global components of RSL. However, these terms represent uniform changes in sea level, which are not equivalent to GMSL changes. The two main causes of GMSL changes are land-mass changes and ocean heat uptake and both give rise to non-uniform patterns of sea-level rise. For this reason, I think that the terminology used here is misleading and it would be more appropriate to call these terms “uniform components” of RSL. Related to this, in lines 118-120 the authors write “Our global sea-level estimate is consistent with other GMSL budgets for the 20th century”. Is your “global sea-level estimate” given solely by the term $g_s(t)$? If yes, I wouldn’t expect such estimate to be consistent with other GMSL since the former is missing important contributions to the global average associated with non-uniform changes. Could you please clarify?

It is correct that we are estimating global sea level via the signal common to all of the records in the Common Era database, following Kopp et al. (2016) and Kemp et al. (2018). In the text, we have generally avoided referring to this “globally uniform” term as GMSL. We did mistakenly imply this in a comparison of our record to tide-gauge based GMSL estimates, and have clarified this in Ln 121. To further clarify that the interpreted global signal is the common uniform term among sites, we added additional clarifiers in Ln 35, 103, 367, and 386.

However, it is incorrect to say that this method would not give an estimate of GMSL in the presence of spatially complete data. There may well be a sampling bias introduced here, but this

approach is not fundamentally different in this regard from some of the methods used to calculate GMSL from tide-gauge data; it can be viewed as a version of the virtual-station method of Jevrejeva et al. (2006).

In addition, consistent with Kopp et al. (2016), the input data for the model also includes the global mean sea-level reconstruction of Hay et al. (2015) from tide-gauge records for 1880-2010. We did an additional run of the model where we removed the Hay et al. (2015) curve from the analysis to see what the method estimates for the common global signal when there is no direct GSL “observation.” In this case, we found that this broadens the 20th century uncertainty, but does not significantly change the mean. For example, in 1900, the global sea level uncertainty is 0.6 cm with the Hay curve and 1.3 cm without the Hay curve; in 2000, the uncertainty is 0.3 cm with the Hay curve and 1.0 cm without.

Line 42. To be precise, the first Argo floats were deployed in 1999 and it was not until 2007 that the program achieved a reasonably good spatial coverage with 3000 floats.

We added a “more recently” clarifier in Ln 43 for this point about the Argo floats.